# Estimating the Risk of Human Herpesvirus 6 and Cytomegalovirus Transmission to Ugandan Infants from Viral Shedding in Saliva by Household Contacts

**DOI:** 10.3390/v12020171

**Published:** 2020-02-03

**Authors:** Bryan T. Mayer, Elizabeth M. Krantz, Anna Wald, Lawrence Corey, Corey Casper, Soren Gantt, Joshua T. Schiffer

**Affiliations:** 1Vaccine and Infectious Diseases Division, Fred Hutchinson Cancer Research Center, Seattle, WA 98109, USA; ekrantz@fredhutch.org (E.M.K.); lcorey@fredhutch.org (L.C.); jschiffe@fredhutch.org (J.T.S.); 2Department of Medicine, University of Washington, Seattle, WA 98195, USA; 3Department of Laboratory Medicine and Microbiology, University of Washington, Seattle, WA 98195, USA; 4Department of Epidemiology, University of Washington, Seattle, WA 98195, USA; 5Infectious Disease Research Institute, University of Washington, Seattle, WA 98102, USA; Corey.Casper@idri.org; 6BC Children’s Hospital Research Institute, Vancouver, BC V5Z 4H4, Canada; sgantt@bcchr.ubc.ca

**Keywords:** HHV-6, CMV, virology, transmission, epidemiology, biostatistics, viral excretion

## Abstract

Human herpesvirus 6 (HHV-6) and cytomegalovirus (CMV) infections are common in early childhood. In a prospective Ugandan birth cohort study, most infants acquired HHV-6 (24/31; 77%) and CMV (20/30; 67%) during follow-up. To assess the transmission risk, we modeled a dose–response relationship between infant HHV-6 and CMV infections and weekly oral viral shedding by mothers and all other (“secondary”) children in the home. Oral viral loads that were shed by mothers and secondary children were significantly associated with HHV-6 but not CMV transmission. While secondary children had higher and more frequent HHV-6 shedding than their mothers, they had a lower per-exposure transmission risk, suggesting that transmission to maternal contacts may be more efficient. HHV-6 transmission was relatively inefficient, occurring after <25% of all weekly exposures. Although HHV-6 transmission often occurs following repeated, low dose exposures, we found a non-linear dose–response relationship in which infection risk markedly increases when exposures reached a threshold of > 5 log_10_ DNA copies/mL. The lack of association between oral CMV shedding and transmission is consistent with breastfeeding being the dominant route of infant infection for that virus. These affirm saliva as the route of HHV-6 transmission and provide benchmarks for developing strategies to reduce the risk of infection and its related morbidity.

## 1. Introduction

Human herpesvirus-6 (HHV-6) infects the vast majority of humans within the first few years of life, and febrile illness (roseola) associated with primary infection is a major cause of health care visits for young children [1,2,3]. Furthermore, HHV-6 leads to encephalitis in immunocompromised hosts and has been implicated as a possible contributor to other central nervous system diseases, including epilepsy and Alzheimer’s disease [4,5,6]. Like other HHVs, HHV-6 establishes latency following primary infection and is persists within the host for life, with intermittent reactivation resulting in the shedding of infectious virus in saliva [7]. 

Cytomegalovirus (CMV) is also widespread globally and commonly infects young children, particularly in the developing world [8,9,10]. CMV is a major cause of illness following organ and stem cell transplantation and during untreated HIV infection [11,12,13,14]. CMV appears to be associated with immune senescence in the elderly, though the long-term consequences of infection are still being determined [15,16,17,18,19].

Because of the public health importance of HHV-6 and CMV infections and the interest in preventing HHV-6-related diseases, understanding the determinants of transmission is of great interest. CMV transmission to infants has been shown to occur most commonly via breast milk, though saliva and urine are other routes of infection during early childhood [9,20,21,22,23]. HHV-6 is likely transmitted through saliva, based on the frequency of the viral detection of the virus in the saliva of infected individuals [3]; in contrast, HHV-6 infection is not associated with breastfeeding and is rarely found in breast milk [9,24]. Thus, HHV-6 transmission is likely determined by the level of exposure to oral, viral shedding by close contacts. Exposure to high quantities of the virus at mucosal sites predicts the transmission of herpes simplex virus type 2 (HSV-2) [25,26] and HIV [27,28]. Estimating the risk of acquisition from the level of exposure to a virus at a mucosal surface has important ramifications for strategies designed to prevent virus transmission [26]. Effective treatment or vaccination strategies might have specific measurable effects on the transmission dose–response curve. Predicting transmission risk is challenging, however, because concurrent data from transmitters and susceptible individuals are required to adequately link exposure and infection. Here, we use longitudinal quantitative viral shedding data within households to estimate the transmission risk of HHV-6 and CMV in a cohort of infants on a weekly basis.

## 2. Materials and Methods 

### 2.1. Study Cohort and Data

Study data were derived from a previously described household-based birth cohort study in Uganda [9]. Pregnant women attending prenatal care at Mulago Hospital in Kampala were eligible if they had at least one other (“secondary”) child <7 years old living in the home and with documented HIV infection status. During the study, home visits were conducted within the first week after birth (median 2 days of age; range 0–9) and each week thereafter. At home visits, oropharyngeal swabs were collected in a standardized manner [29] from the mother and all children. The swabs were collected weekly from infants for up 119 weeks following birth, and from mothers and secondary children in the households for the first 52 weeks. An analysis of the clinical outcomes in this cohort by Gantt et al. [9] found a trend toward fever and increased paracetamol use associated with primary HHV-6 infection and no symptoms associated with CMV infection. 

All oral swabs were tested for HHV-6 and CMV by quantitative (q)PCR, and HHV-6 typing was performed using the previously described methods [30,31,32]. Briefly, DNA extraction was performed using QIAamp DNA Mini Kits (Qiagen, Germantown, MD, USA) according to manufacturer’s instructions. The 5R set of primers and probes were used to amplify HHV-6 [31], and discrimination between HHV-6A and 6B was performed by a separate PCR targeting the U94 gene using species-specific probes [32]. The amplification of CMV was performed using primers and probes that simultaneously target the glycoprotein B (gB, UL55) and immediate early viral protein 1 (IE1, UL123) coding regions, as described [30]. The cutoff for a positive HHV qPCR was 3 viral genome copies/reaction, or ~150 copies/mL of swab buffer [9]. HIV testing was performed according to national guidelines and was negative for all infants [9].

### 2.2. Definitions of Exposure and Transmission Events

We defined a potential salivary shedding exposure as a virus detected in a swab collected from mothers or secondary children when it met the following criteria: 1) it occurred before acquisition in the infant, or 2) it occurred at any time in a household in which no infant acquisition was observed. When there were multiple secondary children in one household measured on the same week, the aggregate sum was used to create a single measurement. In addition to considering the exposure from mother and secondary children independently, we constructed a household measure calculated as the sum of exposures from mothers and the summed secondary children measurements from contemporaneous samples.

While the study visit schedule was designed to be weekly, the actual sampling schedule often deviated from weekly visits. To standardize exposure intervals, the study time was considered relative to infants’ date of birth and one exposure measurement was used per week. If there were multiple measurements per week, we aggregated these values by taking the maximum. If weekly measurements were missing, they were imputed using linear interpolation between the previous and subsequent observed concentrations. In 5 households, exposure data was missing for the first week following infant birth, so linear interpolation could not be performed. For 4 of those households, the first exposure observation occurred during the second week after the infant was born, and this concentration was used to impute the first week exposure. In the fifth household, the first exposure observation was at 7 weeks. This household was included in the descriptive exposure analysis but was excluded from the risk analysis. The interpolation never incorporated concentrations measured after transmission was observed.

The criteria for determining transmission to infants and corresponding infection times were based on the onset of the repeated detection of viral DNA at high levels in oral swabs or in plasma from infants and mothers confirmed with serology for CMV [9]. To estimate the weekly transmission risk, an exposure was considered infectious if it occurred approximately one week prior to detected infection. Because of the limited sample size and variable sampling schedule, we allowed an infectious exposure window around one week, between 4 and 14 days, following an exposure. Infected infants were considered right-censored if there were no infectious exposures measured during this window. For uninfected infants, the infant was considered right-censored after their final observed exposure. The timing of transmission events was displayed using cumulative incidence curves with transmission risk beginning at the date of infant birth and the time to transmission being calculated relative to that date. The cumulative incidence was estimated using the Kaplan–Meier survival estimator assuming viral infections were not competing risks.

### 2.3. Survival Analysis and Dose-Response Modeling

To estimate the weekly risk of transmission by household members, we constructed a time-dependent dose-response model using a survival model. In contrast to a Cox proportional hazards model, where the hazard is not generally estimated, we constructed models by directly estimating the hazard to determine the dose–response relationship between viral load exposure and risk.

As sampling is generally weekly, we assumed that the true transmission time was interval censored and occurred during the week between the last observation prior to infection and the infection time. Therefore, our model was constructed to estimate the weekly transmission risk given weekly measurements of exposures. The resulting weekly infection probability for a given prior weekly exposure (E) was calculated as follows:(1)$$\mathrm{Pr}\left(infection\right)=1-e^{-(b_{0}+b_{E}E)},$$
where b_E_ is the risk parameter associated with the exposure and the b0 parameter is a constant weekly risk parameter capturing the remaining risk not associated with exposure (i.e., when exposure was 0 DNA copies/mL). For the combined model, the weekly infection probability was calculated as follows:(2)$$\mathrm{Pr}\left(infection\right)=1-e^{-(b_{0}+b_{S}S+\;b_{M}M)},$$
where S and M indices correspond to secondary children and mother exposures, respectively. 

The parameters in these models were estimated from the data by maximizing a survival likelihood. Details on the model and model optimization procedures are available in Appendix B. To determine whether household exposures significantly contributed to infection risk, the model incorporating exposure in the household was statistically compared to a null model including just the constant weekly risk parameter, b_0_. This was done using a likelihood ratio test, which is described in more detail in Appendix B. The predictions of the weekly risk were calculated from the individual (Equation (1)) and combined (Equation (2)) model using the fitted parameters using the range of exposures observed in the data. 

An infectious dose of viral load associated with 50% risk (ID_50_) was calculated by solving for the exposure that gives a 50% infection risk given the fitted parameters using Equations (1) and (2). ID_25_ and ID_75_ were also calculated. For the combined model (Equation (2)), ID calculations were done for one exposure source at a time (the other was assumed to be 0 DNA copies/mL).

### 2.4. Sensitivity Analysis

We performed a series of sensitivity analyses. First, we iteratively refit the model, leaving out one household to evaluate individual household leverage on the parameter estimates. Next, to evaluate the sensitivity to exposure interpolation, we iteratively refit the model, allowing an increasing maximum amount of exposure interpolation by households. In other words, we fit the model including households with 20% interpolation or less, then 25% interpolation or less, etc., until all households were included. Lastly, in combination with the interpolation analysis, we also fit the model using only data from households with transmission to assess whether there was a categorical difference that determined whether infants were not infected that could modulate exposure risk.

### 2.5. Software, Data, and Code Availability

Programming and analysis was conducted using the R programming language (CRAN) [33]. Data processing and visualizations were conducted using the tidyverse [34]. Kaplan–Meier estimates were generated using the survival package [35]. Exact Wilcox rank tests were performed using the coin package [36,37]. Optimization was done using the optim function with the BLGS [38] and Nelder–Mead [39] algorithms. Latin hypercube samples were implemented using the lhs package [40]. A research compendium for this analysis was created using the workflowr package [41]. The workflowr package generated the following research website containing all data and analysis code used to generate results and figures: http://bryanmayer.github.io/HHVtransmission. 

## 3. Results

### 3.1. HHV-6 and CMV Transmission Occurred at High Rate Early During Infancy

Thirty-two newborn infants, 32 mothers (17 HIV-infected and 15 HIV-uninfected), and 49 secondary children (4 HIV-infected and 7 unknown) were followed for a median of 57 weeks. Primary HHV-6 infections were observed in 24 of the 31 at-risk infants during the study (one household had no detectable virus during the study). All infections were typed as HHV-6B and 1 was a co-infection with HHV-6B and HHV-6A. CMV was acquired at a similar rate, with 20 infections observed among 30 infants at risk of infection during the study (two cases of congenital CMV transmission were excluded). The majority of transmissions occurred prior to one year with no difference in cumulative incidence observed between infants with HIV+ and HIV- mothers for either virus (Figure 1). As exposure information was only collected through the first year, the four CMV transmissions and one HHV-6 transmission that occurred after one year were right-censored at one year for further analysis.

### 3.2. Secondary Children Shed at Higher Viral Loads than Mothers

Overall, during the at-risk period for the infants, there were 684 exposure weeks for HHV-6 and 819 exposure weeks for CMV to evaluate for transmission. Not every week had a measured exposure; secondary children were more likely to have missing data with 21% observations missing compared to 6% in mothers across the viruses (Appendix A). Observed (non-missing) exposures were summarized for each household contact and virus by the frequency of positive measurements (percent of weeks with positive oral swabs) and then the mean and maximum oral viral loads (log_10_ DNA copies/mL) over the exposure period. For both viruses, secondary children uniformly had higher shedding frequencies and viral loads compared to mothers in the same household (Figure 2a–c). Summarized exposure measurements were not significantly correlated between mothers and secondary children within the same household, although there were modest positive correlations estimated for HHV-6 (Spearman correlation *p*-values > 0.05 for all, Appendix A). Overall household exposure was assessed by taking the sum of maternal and secondary children viral loads at each week. In the majority of households, viral loads shed by secondary children contributed to the majority of the household exposure sum measure, comprising a median of 99.7% (IQR: 94.6–99.9%) of the CMV household sum and 91.2% (IQR: 80.5–98.8%) of the HHV-6 household sum (Appendix A).

### 3.3. Correlation between household shedding and transmission to infants

We next compared shedding frequency, median viral loads, and maximum viral loads between households with (prior to) and without a transmission event for each virus (Figure 2d–f, Appendix A). Generally, for both HHV-6 and CMV, households with transmission events had higher oral shedding frequencies and magnitudes than households without observed transmission in the first year. For HHV-6, this trend was uniformed and more evident. For all three outcomes, secondary children and overall household exposures were significantly higher in the households with HHV-6 transmissions (exact Wilcoxon test *p*-values < 0.05, Appendix A). There were no significant differences between shedding patterns from any exposure source between households with and without infant CMV transmission (Appendix A).

### 3.4. HHV-6, but not CMV, Acquisition in Infants was Associated with Viral Loads of Oral Shedding Exposures

We next estimated the weekly risk of transmission by household members using the longitudinal, weekly exposure data. Based on a sensitivity analysis, one household was removed from the HHV-6 transmission model, see Appendix C. All exposure sources were significantly associated with infant HHV-6 infection but not CMV (Table 1). The weekly risk was calculated based on the model estimates and the mean viral load (among positive swabs), using secondary children and mothers by virus as inputs. Without including exposure, the average (constant) weekly risk for HHV-6 was 3.60% and it was 1.96% for CMV. In the HHV-6 models including exposure sources, the constant weekly risk was lower because the exposure sources explain additional variation in weekly risk (Table 1). For secondary children, the estimated weekly risk absent exposure was 2.07% compared to 2.92% given the exposure of the mean positive oral viral load. For mothers, the estimated weekly risk absent exposure was 2.88% compared to 3.39% given as a mean exposure. For either household contact, risk was substantial (> 90%) given the exposure of the maximum observed viral load. As secondary children viral loads comprise a large proportion of household sum exposure, risk due to household sum was similar to secondary children. For CMV, the estimated constant weekly risk in models including exposure sources was 1.92% and did not differ based on the exposure source. With the exception of the household removed from the HHV-6 model, model results were generally robust to sensitivity analysis; for further details and estimated model parameters (Table A1), see Appendix C.

### 3.5. A Combined Exposure Model for HHV-6 Highlights Differential Risk between Mother and Secondary Children Exposures

While the household sum model is a model of combined household risk, it assumes that risk exposure is equivalent between secondary children and mothers. We next estimated a combined exposure transmission risk that includes and adjusts for separate risk estimates by household secondary children and mothers. The combined model for HHV-6, but not CMV, was statistically significant (Table 2). In the combined model, the estimated constant weekly risk was 1.80%, which was lower than any estimate from the individual models (Table 1), indicating that additional risk is explained by including both exposures simultaneously with a different risk. Per mean exposure, absent exposure from another contact source, and secondary children exposure had a higher weekly risk compared with maternal exposure: 2.55% vs. 2.22% a result following from higher viral loads in secondary children (Figure 2).

For equivalent viral loads, the risk of HHV-6 infection was higher from maternal exposures. This is evident by higher maximum maternal exposure risk despite lower maximum viral load (Table 1 and Table 2) and differences in infectious doses resulting in a given risk of infection (Table 3). From the combined exposure model, the estimated infectious oral exposure, resulting in 50% probability of infection (ID_50_), was 5.01 log_10_ DNA copies/mL from maternal exposures compared to 5.92 log_10_ DNA copies/mL from secondary children, which is almost a 10-fold difference. Using the single exposure models, ID_50_ estimates showed a similar difference between sources: 4.92 and 5.87 log_10_ DNA copies/mL from mother and secondary children, respectively. As exposure to CMV was not associated with infection risk, ID calculations were not performed for CMV.

### 3.6. The Dose–Response Relationship between the Viral Load of HHV-6 oral Shedding Exposure and Weekly Infant Acquisition

Using the estimated transmission models, we can describe the dose–response relationship by calculating the probability of transmission per weekly exposure across the range of potential exposures. Comparing the estimated dose-response curve from the combined exposure model to the corresponding observed data, a clear dose–response relationship was evident between weekly oral HHV-6 shedding and weekly transmission (Figure 3a). In general, the probability of transmission on any given week was relatively low, with the majority of individual exposures having a < 25% probability and only high exposures (> 5 log_10_ DNA copies/mL for mothers or > 6 log_10_ DNA copies/mL for secondary children) result in higher predicted risk. For CMV, as there was no risk attributable to exposure, the dose–response curves were flat, always predicting the 2% constant risk estimated from the model (Appendix A).

We next looked at the weekly risk resulting from combined exposures from both secondary children and mother within the household. Most weekly exposures were either low without observed transmission (less than the lower limit of detection) or in the 3-5 log_10_ DNA copies/mL range where the majority of transmissions were observed (Figure 3b). Of note, there were 107 exposure weeks with no measured household HHV-6 shedding measured and no instances of HHV-6 infections. In contrast, for CMV, there were 4 infections recorded among the 217 weeks with no household CMV shedding (Appendix A). Using the model, the majority of combined exposures were associated with a low weekly probability of HHV-6 transmission (Figure 3c). The predicted risk increases dramatically for HHV-6 exposures higher than 5.5 log_10_ DNA copies/mL from secondary infants and 4.5 log_10_ DNA copies/mL from the mother. Though these high exposures were generally rare in the data, they were associated with an increased observed transmission risk, particularly when the mother had a detectable viral load. Together, the model and the data suggest that the majority of HHV-6 infections arise from repeated exposures to oral viral load shedding by all members in the household, but with varying degrees of contact efficiency. Episodes of high viral shedding by any household member substantially increases risk.

## 4. Discussion

Here, we estimated risk of HHV-6 and CMV transmission to infants using weekly measurements of oral shedding exposures by household contacts. Oral viral shedding by contacts was strongly associated with incident HHV-6 infection in infants, which affirms suggestions that saliva is a major transmission route [3,24] and is consistent with the epidemiologic data showing that parental saliva-sharing behavior and having older siblings are both associated with HHV-6 acquisition risk [42]. HHV-6 transmission showed a clear dose–response relationship with weekly exposure to oral viral loads from both mothers and secondary children, with different associated risks. Infection generally resulted from repeated, low infectivity exposures. However, the relationship between viral load and transmission appears non-linear and the data suggest that exposure to bursts of oral shedding with viral loads >1 million copies dramatically increases the risk of transmission. 

We did not find that oral exposure was a significant risk for infant CMV acquisition, which is consistent with breast milk being the dominant route in this age group or this infant cohort specifically [9]. Although oral shedding by household contacts did not appear to be relevant for infant acquisition of CMV in this setting, we posit that there is a relationship between the CMV viral load of an exposure and transmission. We did not collect breast milk from mothers in this cohort, but others have reported a positive correlation between the viral load in breast milk and the risk of infant CMV acquisition [22,43,44], and we have shown a dose–response between oral CMV viral load shedding by young children and transmission to their mothers [45]. Breast milk does not appear to be a route of HHV-6 transmission [24]; rather, saliva has been thought to be the major source of HHV-6 transmission based on the frequency of viral detection in saliva [3], which our data strongly support. 

While our model estimated the weekly risk from a single oral viral load measurement, there are likely many exposures in a given week that vary in magnitude. In contrast to our approach, a mechanistic dose–response relationship would describe the probability of infection at each exposure. Data informing such a model would be extremely difficult, if not impossible, to collect. If weekly measurements are a good estimate of the average daily exposures within a week, then we expect the true risk per contact to be lower than our weekly risk estimate because there are likely many exposures per week. Specifically, while mothers have lower viral loads, they likely expose their infants more than siblings, and therefore may have similar per exposure risks. In this case, infants become high-risk when their mothers have a high shedding event due to exposure frequency, which could exponentially increase transmission risk even amongst low risk, individual exposures. Regardless, this is consistent with our conclusion that HHV-6 transmission may result from repeated, inefficient exposures. Ultimately, with increased exposure sampling granularity, a true per-contact risk could be estimated with this model approach and has been previously shown in HSV-2, a system with discrete, sexual contacts [26].

Time-dependent exposure and transmission data are challenging to collect, and our analysis was therefore limited by a small sample size and censoring. To accommodate, we used interpolation for missing exposures, interval censoring approaches to model survival risk, an independent weekly risk assumption, and a linear risk model structure within the survival function. In addition, all HHV-6 positive samples could not be typed and so the exact proportion that were HHV-6B is unknown. These approaches result in assumptions that add uncertainty to our model results that are difficult to quantify. Reassuringly, we found that the oral shedding of HHV-6 by household members is a dose-dependent risk factor for transmission with or without data interpolation. While our estimated dose–response relationship for HHV-6 is consistent with these data and the known transmission routes of HHV-6, the validation of the relationship requires additional study.

Primary HHV-6 infection accounts for 10%–20% of febrile illness in the first 2-3 years of life in high-income countries and is associated with a high rate of physician visits [3,7]. Thus, the prevention of HHV-6 infection is of significant public health interest. Linking HHV-6 exposure to infection risk provides insight into the viral determinants of transmission. In general, the classification of a subpopulation as high- or low-risk depends on both the frequency of exposure and the susceptibility to infection. There is no known immunologic correlate of protection against primary infection with HHV-6, which infects nearly 100% of people worldwide. Using the dose–response relationships and the subsequent infectious doses (e.g., ID_50_) uncovered in this study, we established a baseline link between oral shedding and the weekly risk of transmission. Interventions that affect the susceptibility of infants (i.e., vaccination) would theoretically increase necessary quantities of exposure to result in infant infection, while antiviral therapy for infected contacts could lower exposure. The study of dose–response relationships before and after specific interventions could be used as a precise metric to develop and evaluate the effectiveness of interventions to prevent HHV-6 infection and its related morbidity.

## Figures and Tables

**Figure 1 viruses-12-00171-f001:**
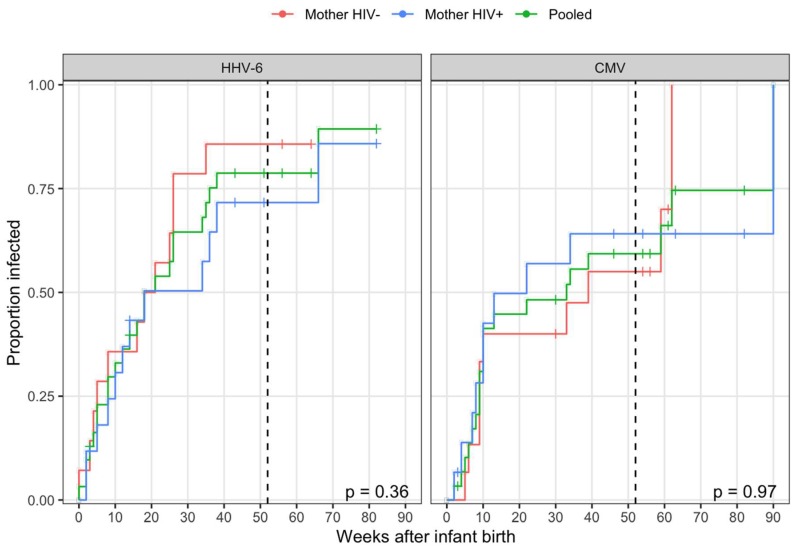
Time taken for the transmission of human herpesvirus 6 (HHV-6) and cytomegalovirus (CMV). Cumulative incidence curves for all transmission events to infants following birth stratified by maternal HIV status with censoring denoted by a plus. The P-values from log-rank tests comparing survival curves (one minus proportion infected) between infants with HIV+ and HIV- mothers are displayed in the bottom-right of each panel. The household exposure information was collected through one year (52 weeks; vertical, dashed black line). All events occurring after one year were right-censored for an estimation of transmission risk.

**Figure 2 viruses-12-00171-f002:**
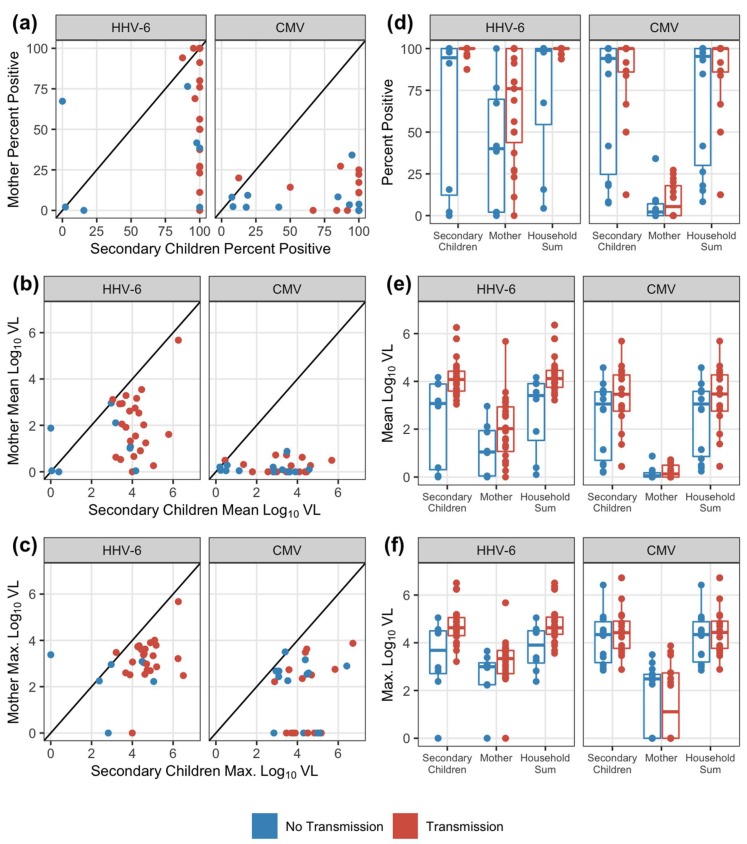
Comparisons of exposure patterns across households and among household members. Comparing viral exposure patterns for HHV-6 and CMV between mother and secondary children within a household by (**a**) percent positive weekly swab measurements; (**b**) mean log viral load (VL; DNA copies/mL); and (**c**) maximum log viral load. The diagonal line in the scatterplot denotes equivalent summary measures between household mother and secondary children (i.e., the points below the line indicate lower measurements from the household mother). Shedding patterns by exposure source compared between households without and with transmission by (**d**) the percent of positive weekly measurements; (**e**) the mean log viral load; and (**f**) the maximum log viral load. Boxes represent the interquartile range, stems represent values within 1.5 of the IQR, and points represent the raw data. All observed exposures measured in the infant’s first year of life and prior to transmission times are shown.

**Figure 3 viruses-12-00171-f003:**
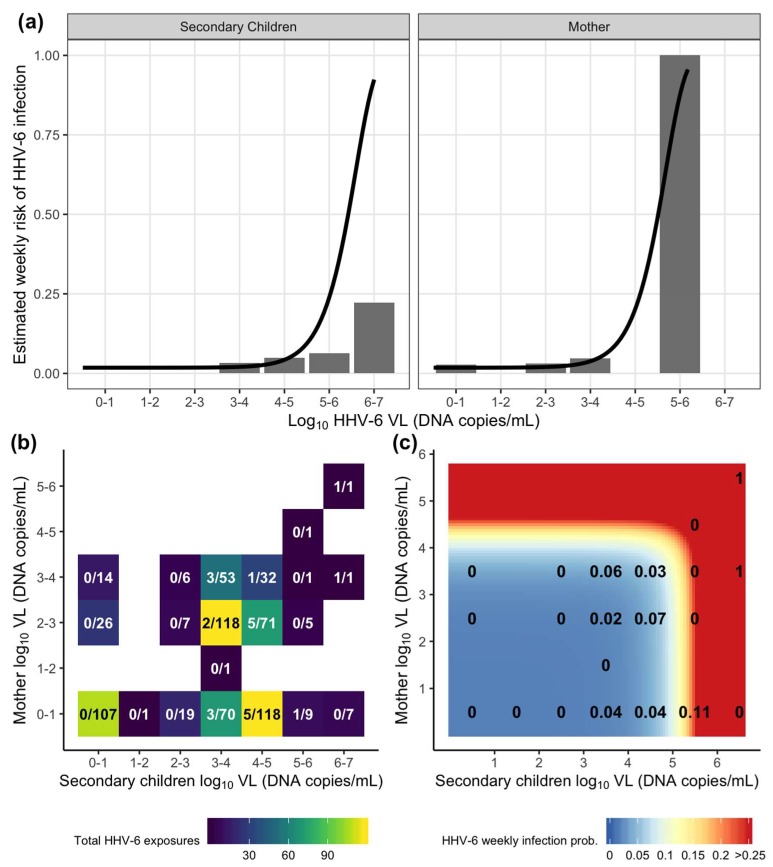
Estimation of HHV-6 acquisition risk from exposures by week using the combined exposure model (Table 2 and Table A1). (**a**) Individual HHV-6 dose–response relationship for mother and secondary children oral viral load (VL, DNA copies/mL) exposures where bars depict the percentage of infections observed among total exposures in the data and lines depict estimated risk from the model. (**b**) A heatmap depicting the distribution of total combined exposures in a given viral load bin from mothers and secondary children in a household. The text depicts the fraction of total infections over total exposures. (**c**) A heatmap depicting the risk of HHV-6 infection from combined household mother and secondary children exposures estimated from the model. The text depicts the proportion of infections observed in the data using binned viral loads from (b) (equivalent to fraction displayed).

**Table 1 viruses-12-00171-t001:** Estimates of the weekly risk for each virus with no exposure (constant weekly risk) and by exposure source. Models of risk were fit without exposure (null model) and independently for each exposure source. Weekly risk with exposure includes constant risk from the model. The mean exposure was calculated among positive swabs. The P-values were calculated from the model to test if risk attributable to exposure is significantly different from null model without exposure risk (log-likelihood ratio test).

		No Exposure	Mean Exposure		Maximum Exposure		
Virus	Exposure Source	Weekly Risk (%)	Log_10_ DNA copies/mL	Risk (%)	Log_10_ DNA Copies/mL	Risk (%)	*p*-Value
HHV-6	Null	3.60					
	Secondary children	2.07	3.98	2.92	6.50	94.66	<0.001
	Mother	2.88	2.82	3.39	5.68	97.75	0.007
	Household sum	2.01	3.94	2.79	6.50	94.63	<0.001
CMV	Null	1.96					
	Secondary children	1.92	3.72	1.92	6.72	1.92	0.939
	Mother	1.92	2.56	1.92	3.88	1.92	0.939
	Household sum	1.92	3.77	1.92	6.72	1.92	0.939

**Table 2 viruses-12-00171-t002:** Estimates of weekly risk for each virus with no exposure (constant weekly risk) and by exposure source. Models of risk were fitted to adjust for both exposure sources and risks, which were calculated assuming that other exposure sources were 0 DNA copies/mL. The weekly risk with exposure includes constant risk from the model. P-values calculated from the model to test if risk attributable to exposure is significantly different from the null model without exposure risk (log-likelihood ratio test).

	No Exposure	Secondary Children Exposure ^1^		Maternal Exposure ^1^		
Virus	Weekly Risk (%)	Mean Exposure Risk (%)	Maximum Exposure Risk (%)	Mean Exposure Risk (%)	Maximum Exposure Risk (%)	*p*-Value
HHV-6	1.80	2.55	92.39	2.22	95.58	<0.001
CMV	1.92	1.92	1.92	1.92	1.92	0.997

^1^ Mean and maximum exposures (DNA copies/mL) shown in Table 1.

**Table 3 viruses-12-00171-t003:** HHV-6 weekly infectious exposure (ID, log_10_ DNA copies/mL) associated with 25%, 50%, and 75% probability of infection (ID_25_, ID_50_, and ID_75_) estimated from the HHV-6 risk models by exposure source. For the exposure combined model, calculations assume a single exposure source. Exposure was not associated with CMV transmission and so IDs were not calculated.

		HHV-6 Infectious Dose (ID, log_10_ DNA copies/mL)		
Model	Exposure Source	ID_25_	ID_50_	ID_75_
Individual	Secondary Children	5.47	5.87	6.17
	Mother	4.51	4.92	5.23
Combined	Secondary Children	5.53	5.92	6.23
	Mother	4.61	5.01	5.32

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
