# Peer review of "Estimating the Risk of Human Herpesvirus 6 and Cytomegalovirus Transmission to Ugandan Infants from Viral Shedding in Saliva by Household Contacts"

_viruses, 2020, doi:10.3390/v12020171_

Round 1
Reviewer 1 Report
Mayer et al have investigated the risk of HHV-6 and CMV transmission to infants from viral shedding in saliva by mother and other children in the home. They found that oral viral loads sheds by mother and secondary children were significantly associated with HHV-6 transmission, and transmission to maternal contacts may be more efficient. The number of enrolled infants were relatively small, but this study represents a massive effort, and the results are convincing. Specific comments are as follows;
In this study, infants with young siblings were enrolled, and authors have concluded that saliva of both mother and siblings would be a major transmission route of HHV-6. If oral viral shedding by siblings plays an important role in HHV-6 transmission, the timing of HHV-6 transmission in the first child should be later than other siblings. Are there any published data regarding this point?
Swabs were collected for up 119 weeks following birth. Did enrolled infants have opportunities to attend daycare facilities during that period? It would be possible that HHV-6 or CMV had transmitted from other children with close contact.
It would be quite important that transmission of HHV-6 or CMV was symptomatic or not. It should be shown whether clinical symptoms such as fever or skin rash were accompanied by viral transmission. Furthermore, it would be interesting if viral loads of infants were associated with clinical symptoms.
Author Response
Mayer et al have investigated the risk of HHV-6 and CMV transmission to infants from viral shedding in saliva by mother and other children in the home. They found that oral viral loads sheds by mother and secondary children were significantly associated with HHV-6 transmission, and transmission to maternal contacts may be more efficient. The number of enrolled infants were relatively small, but this study represents a massive effort, and the results are convincing. Specific comments are as follows;
We are greatly appreciative of the reviewer's time and effort reviewing our manuscript.
In this study, infants with young siblings were enrolled, and authors have concluded that saliva of both mother and siblings would be a major transmission route of HHV-6. If oral viral shedding by siblings plays an important role in HHV-6 transmission, the timing of HHV-6 transmission in the first child should be later than other siblings. Are there any published data regarding this point?
This is an excellent question. Indeed, there are epidemiologic data to support this, this has now been referenced in the discussion (Rhoads et al, J Infect 2007 [42]) (Lines 314-315).
Swabs were collected for up 119 weeks following birth. Did enrolled infants have opportunities to attend daycare facilities during that period? It would be possible that HHV-6 or CMV had transmitted from other children with close contact.
This is possible. We did not collect information about daycare, which seemed to be quite uncommon among the families we followed in Kampala. However, contact with other children (neighborhood, extended family, etc.) was likely frequent, and this is certainly a possibility that we cannot exclude.
It would be quite important that transmission of HHV-6 or CMV was symptomatic or not. It should be shown whether clinical symptoms such as fever or skin rash were accompanied by viral transmission. Furthermore, it would be interesting if viral loads of infants were associated with clinical symptoms.
The associations between acquisition of these infections and symptoms of illness were reported in an earlier publication (Gantt JID 2016). Primary HHV-6 infection showed a trend toward fever and paracetamol use, and CMV was not associated with any symptoms. We added a sentence in the methods pointing to this analysis (Lines 70-72). However, as discussed in that paper, the numbers of acquisition events was small, and the background rate of symptoms of illness was extremely high; fever, diarrhea, rash and cough occurred in all infants during the study, irrespective of infection status. This likely limited our ability to discern HHV-6-related symptoms, which have been well described in higher-income settings (e.g., Zerr NEJM 2005 [3]). Other studies have shown a relationship between HHV-6 viral load and symptoms of illness, both in otherwise healthy children (https://adc.bmj.com/content/77/1/42) and in immunocompromised patients (https://www.ncbi.nlm.nih.gov/pmc/articles/PMC5820136/).
Reviewer 2 Report
Mayer and co. present here data from a well-conducted cohort study on 32 newborn infants, showing a risk model for HHV-6 transmission based on data from viral shedding in saliva by the mother and other household members
The paper is well-written and the methodology is clear as well as the presentation of data and related discussion. I only have two minor comments:
1) in the abstract, last sentence, I would add "to reduce the risk of infection and its related morbidity", in order to be more consistent with the Discussion: indeed, since the vast majority of HHV-6 infections are benign, strategies to prevent them are useful for specific population at risk or in those situations where complications may be expected, increasing morbidity
2) is mother-related HHV-6 infection influenced by concurrent presence of virus among other household members? In other words, did author compare the risk of HHV-6 infection, when it occured presumably from the mother, in the presence or absence of shedding within saliva of other members? This would answer to a possible sinergistic effect (or none) played by concomitant presence of virus in mother and any of the other household members
Author Response
Mayer and co. present here data from a well-conducted cohort study on 32 newborn infants, showing a risk model for HHV-6 transmission based on data from viral shedding in saliva by the mother and other household members
The paper is well-written and the methodology is clear as well as the presentation of data and related discussion. I only have two minor comments:
Thanks so much for this comment and taking the time to review our manuscript.
1) in the abstract, last sentence, I would add "to reduce the risk of infection and its related morbidity", in order to be more consistent with the Discussion: indeed, since the vast majority of HHV-6 infections are benign, strategies to prevent them are useful for specific population at risk or in those situations where complications may be expected, increasing morbidity
This has been added as suggested.
2) is mother-related HHV-6 infection influenced by concurrent presence of virus among other household members? In other words, did author compare the risk of HHV-6 infection, when it occured presumably from the mother, in the presence or absence of shedding within saliva of other members? This would answer to a possible sinergistic effect (or none) played by concomitant presence of virus in mother and any of the other household members
This is an interesting suggestion but is difficult to assess. To address the reviewer’s suggestion, one would need to isolate the per exposure contact risk, which unfortunately cannot be done with our data because our exposure risk is a composite of both exposure frequency per week and per exposure contact risk due to the weekly sampling scheme. We also would require households without secondary children to establish a maternal risk baseline. Given our model identified risk signal specific to maternal exposure patterns, we expect the main results of the paper relying on additive risk to hold whether or not there is some interaction/synergy. Exposure independence (additive risk) is a standard assumption of dose-response modeling partially because it's generally difficult to relax this assumption (see Mayer et al. RSIF 2010 [46] for discussion of experimental design and sample size required to assess synergy via temporal exposure dependence).
Reviewer 3 Report
Mayer et al have presented an analysis of the risk of HHV-6 and CMV transmission in Ugandan infants from oral viral shedding within the household. The authors have utilised a previously recruited prospective birth cohort study to provide the data which was used to model and assess the transmission risk. They have clearly presented the results from the study demonstrating transmission of both pathogens to infants within the first year of birth and broken down the exposure to infants due to oral shedding of HHV6 and CMV between mothers and siblings present in the household. They have shown that exposure to oral shedding of HHV6 is significantly associated with transmission to infants but CMV transmission is not associated with oral shedding. The authors also modelled the risk of infection of HHV6 and concluded that both the transmission model and the raw data suggests that the majority of HHV6 infections in infants arise from repeated exposure to low level oral shedding of HHV6.
Overall this paper is well written, and the complex statistical modelling used carefully explained in both the main paper and associated appendices. I do however feel that both the introduction and discussion would benefit with some expansion as to why it is important to understand the rates of transmission of HHV6 in infancy is an important public health and clinical concern.
I also noted (I think) some errors in the referencing of the methods for the generation of the qPCR data (Section 2 – study cohort and data section line 63) and feel that firstly a brief summary of the main method used to extract DNA from the oral swabs would be helpful particularly as this is not outlined in the reference used at this point of the methods (from Johnston et al 2009 #29). Secondly a brief description of the targets used for the CMV and HHV6 qPCR would be also be helpful, particularly as the listed references (30 – 32) contain several different primer/probe target options for both viruses. Also, I did not see in references 9, 30 – 32 a method for HHV6 A or B typing, although from my literature search, I see that some of the same authors have published this method in other papers. I mention this as I feel anyone interested in how the data for HHV6 and CMV from oral swabs were quantified and typed would not be able to find this easily from this manuscript and would have to invest a lot of time chasing down references within references to find out how this was achieved. However, as it has all been published a brief summary of the pertinent facts should be enough.
Author Response
Mayer et al have presented an analysis of the risk of HHV-6 and CMV transmission in Ugandan infants from oral viral shedding within the household. The authors have utilised a previously recruited prospective birth cohort study to provide the data which was used to model and assess the transmission risk. They have clearly presented the results from the study demonstrating transmission of both pathogens to infants within the first year of birth and broken down the exposure to infants due to oral shedding of HHV6 and CMV between mothers and siblings present in the household. They have shown that exposure to oral shedding of HHV6 is significantly associated with transmission to infants but CMV transmission is not associated with oral shedding. The authors also modelled the risk of infection of HHV6 and concluded that both the transmission model and the raw data suggests that the majority of HHV6 infections in infants arise from repeated exposure to low level oral shedding of HHV6.
Overall this paper is well written, and the complex statistical modelling used carefully explained in both the main paper and associated appendices.
We thank the reviewer for this summary and feedback and greatly appreciate the time and effort they have expended to help improve our manuscript.
I do however feel that both the introduction and discussion would benefit with some expansion as to why it is important to understand the rates of transmission of HHV6 in infancy is an important public health and clinical concern.
We agree that emphasizing the importance of HHV-6 infection is worthy. In the introduction, we think that this is already well addressed: “Human herpesvirus-6 (HHV-6) infects the vast majority of humans within the first few years of life, and febrile illness (roseola) associated with primary infection is a major cause of health care visits for young children [1-3]. Furthermore, HHV-6 leads to encephalitis in immunocompromised hosts and has been implicated as a possible contributor to other central nervous system diseases including epilepsy and Alzheimer’s disease [4-6].” (Lines 34 - 38). In addition, the following has now been added to the discussion: “Primary HHV-6 infection accounts for 10-20% of febrile illness in the first 2-3 years of life in high-income countries, and are associated with a high rate of physician visits (Zerr NEJM 2005 [3], Agut Clin Micro Rev 2015 [7]). Thus, the prevention HHV-6 infection is of significant public health interest.” (Lines 355 - 357).
I also noted (I think) some errors in the referencing of the methods for the generation of the qPCR data (Section 2 – study cohort and data section line 63) and feel that firstly a brief summary of the main method used to extract DNA from the oral swabs would be helpful particularly as this is not outlined in the reference used at this point of the methods (from Johnston et al 2009 #29). Secondly a brief description of the targets used for the CMV and HHV6 qPCR would be also be helpful, particularly as the listed references (30 – 32) contain several different primer/probe target options for both viruses. Also, I did not see in references 9, 30 – 32 a method for HHV6 A or B typing, although from my literature search, I see that some of the same authors have published this method in other papers. I mention this as I feel anyone interested in how the data for HHV6 and CMV from oral swabs were quantified and typed would not be able to find this easily from this manuscript and would have to invest a lot of time chasing down references within references to find out how this was achieved. However, as it has all been published a brief summary of the pertinent facts should be enough.
Thank you for this comment. We apologize for the lack of clarity, and have now specified the extraction method, the method for HHV-6 typing, and all of the primers and probes used for both viruses, with the relevant references (see Lines 74-80).
Round 2
Reviewer 1 Report
The manuscript has been revised well. I think this manuscript will be acceptable.